# Learning Differentiable Tensegrity Dynamics using Graph Neural Networks

**Nelson Chen[1], Kun Wang[2], William R. Johnson III[3],**
**Rebecca Kramer-Bottiglio[3], Kostas Bekris[1], Mridul Aanjaneya[1]**
[1]Rutgers University, [2]Amazon Robotics, [3]Yale University

**Abstract:** Tensegrity robots are composed of rigid struts and flexible cables. They constitute an emerging class of hybrid rigid-soft robotic systems and are promising systems for a wide array of applications, ranging from locomotion to assembly. They are difficult to control and model accurately, however, due to their compliance and high number of degrees of freedom. To address this issue, prior work has introduced a differentiable physics engine designed for tensegrity robots based on first principles. In contrast, this work proposes the use of graph neural networks to model contact dynamics over a graph representation of tensegrity robots, which leverages their natural graph-like cable connectivity between end caps of rigid rods. This learned simulator can accurately model 3-bar and 6-bar tensegrity robot dynamics in simulation-to-simulation experiments where MuJoCo is used as the ground truth. It can also achieve higher accuracy than the previous differentiable engine for a real 3-bar tensegrity robot, for which the robot state is only partially observable. When compared against direct applications of recent mesh-based graph neural network simulators, the proposed approach is computationally more efficient, both for training and inference, while achieving higher accuracy. Code and data are available at
https://github.com/nchen9191/tensegrity_gnn_simulator_public

**Keywords:** tensegrity robots, graph neural networks, differentiable simulation

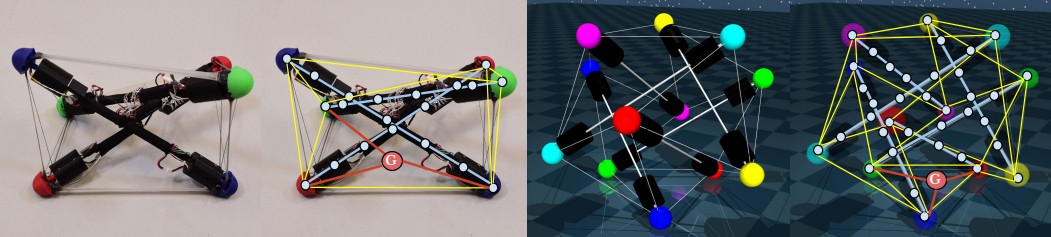

Figure 1: (Left) A real 3-bar tensegrity robot and the graphical representation superimposed. (Right) A simulated 6-bar tensegrity robot in MuJoCo and the graphical representation superimposed. For both platforms, the graph consists of nodes (white) along the rods' axes connected with body edges (blue). Cable edges (yellow) connect two nodes on end caps of different rods. A special ground node (red G node) has contact edges (red) connected to body nodes close to the ground.

## 1 Introduction

Tensegrity robots consist of rigid struts (rods) and flexible elements (cables). This allows them to be lightweight while exhibiting both compliance and rigidity. They can locomote by changing their shape via actuation of their cables. Tensegrity robots are attracting interest due to their wide array of possible applications, such as manipulation [1], locomotion [2], morphing airfoils [3], and spacecraft landing [4]. Recent work [5] has also explored the assembly of multiple tensegrity robots to form and perform complex structures and tasks. Tensegrity robots, however, are difficult to accurately model and control due to their high number of degrees of freedom, significant nonlinearities, and complex dynamics, involving oscillatory behaviors and compliant mechanisms.

This work proposes the use of graph neural networks (GNNs) to improve on previous work on the modeling of tensegrity robots using differentiable engines [6, 7, 8], the current state of the art

8th Conference on Robot Learning (CoRL 2024), Munich, Germany.

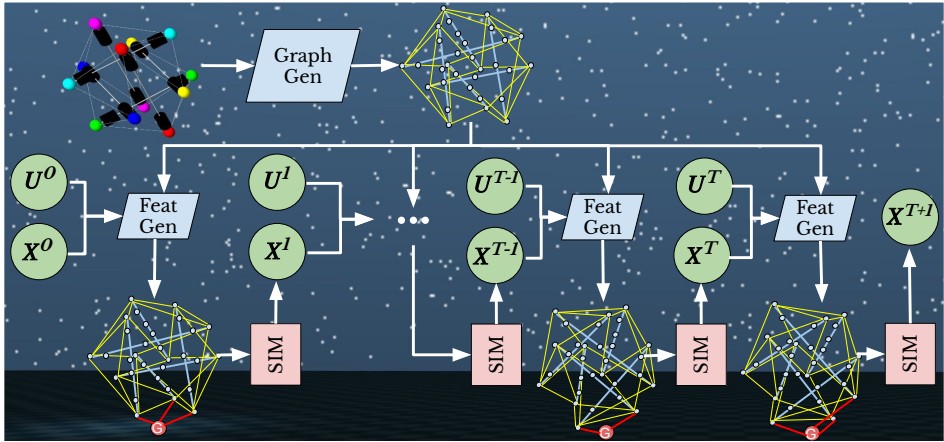

Figure 2: A trajectory rollout: Upon initialization, the tensegrity graph structure is generated and then used as input at every simulation step - a state $\mathbf{X}^t$ and controls $\mathbf{U}^t$ are passed to a feature generation module that updates the graph structure with contact edges and generates feature vectors for all nodes $\{N_i\}$ as well as edges $\{E_{ij}\}$. These are passed to the simulator to predict $\mathbf{X}^{t+1}$.

in matching simulation to reality for tensegrity robots. Inspired by recent efforts that demonstrate GNNs' success in modeling discontinuous, rigid-body contact [9, 10, 11], a GNN-based framework is proposed for learning contact mechanics for tensegrity robots. The method injects a structural prior by representing tensegrities as graphs. These graphs are constructed by leveraging the robots' natural graph-like connectivity with cables between rod end caps, as well as decomposition of the rigid rods to smaller primitive objects (e.g., spheres and cylinders), which constitute object-level nodes (Fig. 1). This graph representation has orders of magnitude fewer nodes and edges than the mesh-based representation introduced in prior GNN work for rigid-body simulators [9], facilitating faster training and inference with less hardware and fewer computational resources, while achieving higher accuracy.

The accompanying experiments evaluate the method and compare it against the first-principles differentiable physics engine in simulation and for a real tensegrity robot. In simulation, MuJoCo [12] provides the ground-truth system. The comparison shows that the learned GNN model performs better than the baseline, for both the 3-bar and 6-bar tensegrity variants, when it has full observability of the robot's state. The available data for a real robot correspond to captured trajectories on a 3-bar tensegrity using a vision-based solution [13]. The real data, however, only include end cap positions of the rods, but without instantaneous velocities at each time step, which are critical for simulating active motion. Due to this partial observability, the first-principles differentiable engine [8] can only be trained with the whole trajectory where the initial state is at rest. In contrast, the GNN model does not require instantaneous velocity information and can improve upon the baseline by warm-starting with simulation data first and then fine-tuning given the real data. The evaluation includes measuring the computational requirements of mesh and surface-based versions of GNNs used in prior work [9, 10, 11] compared to the proposed sparser representation. The proposed method requires less computational resources for both training and inference. Finally, an ablation study evaluates the different choices of the proposed pipeline. In summary, the key contributions are:

- A simple and computationally efficient graph representation for tensegrity systems using object-level nodes and edges that can be used for modeling them using a GNN.
- A learning pipeline that combines both analytical physics components and a GNN to train models that are predictive and stable over long rollouts.
- Evaluations, in simulation and reality, of the GNN in modeling tensegrity robot dynamics.

## 2 Related Work

One family of **tensegrity simulators** is based on solving systems of analytical differential equations representing tensegrity dynamics and structures. These include TensegrityMATLABObjects [14], Software for Tensegrity Dynamics [15], and Models of Tensegrity Structures [16]. These simulators

are built using MATLAB and only support frictionless contact. Other simulators support frictional contact and are based on traditional engines where governing equations from physics, analytical models, and numerical solvers are used. Paul et al. [17] built a simulator on top of ODE [18], which supports massless and volumeless virtual cables but has no contact. The open-source NASA tensegrity robot toolkit [19] and Caliper [20] are built on top of Bullet [21]. These simulators can still suffer from a large simulation-to-reality gap due to unknown system parameters or unmodeled physics. To address this, Wang et al. developed a differentiable physics engine for tensegrity robots [6, 7, 8], which serves as the baseline for this work.

**Differentiable physics engines** have been gaining momentum in robotics. They are able to identify and learn model parameters via gradient-based optimization, often resulting in faster training and better data efficiency. These simulators range from fully first-principled [22, 23, 24, 25, 26] to purely data-driven [27, 28, 29, 30, 31], as well as hybrid setups [32, 33, 34].

One emerging data-driven modeling choice is **graph neural networks** (GNNs). GNNs are able to model particle-based physics [35, 36, 37, 38], mesh-based physics [39, 40, 41], and deformable rod dynamics [42]. Contrary to prior works [33, 43] that suggest that neural networks cannot learn discontinuous, rigid-body contact, recent work [9, 10, 11] demonstrates that GNNs can. Applications in robotics include GNNs modeling object-object and object-gripper interactions [44, 45, 46] and predicting the next states of a soft gripper [47]. Beyond simulation, GNNs have also been used in a variety of other applications, such as modeling robot kinematics [48, 49], multi-robot coordination [50], and path-planning [51]. This work uses a hybrid model with analytical models of actuation and passive forces combined with a data-driven GNN model for ground contact.

## 3   Approach

Tensegrity robots are typically treated as a set of rigid rods and a set of cables that connect between the rods' end caps, referred to as their system topology. The robot state $\mathbf{X}^t$ includes the state of each rod $i$ at time step $t$, $\mathbf{X}_i^t = (\mathbf{P}_i^t, \mathbf{R}_i^t, \mathbf{V}_i^t, \mathbf{\Omega}_i^t)$, consisting of the position $\mathbf{P}_i^t$, orientation $\mathbf{R}_i^t$, linear velocity $\mathbf{V}_i^t$, and angular velocity $\mathbf{\Omega}_i^t$. Then, the simulator can be seen as a function $\mathcal{F}_{SIM}(\mathbf{X}^t, \mathbf{U}^t)$ that takes the current state $\mathbf{X}^t$, along with a set of controls $\mathbf{U}^t$, to predict the next state, i.e.,

$$\mathbf{X}^{t+1} = \mathcal{F}_{SIM}(\mathbf{X}^t, \mathbf{U}^t) \tag{1}$$

The proposed simulator $\mathcal{F}_{SIM}$ uses a GNN to model the robot dynamics where the tensegrity robot is represented by a graph, $\mathcal{G}^t$, with nodes $\mathcal{N}^t$ and edges $\mathcal{E}^t$, at time step $t$. This results in a learnable simulator $\mathbf{GNN}^\theta$ that predicts changes in velocity from contact:

$$\mathcal{G}^t = (\mathcal{N}^t, \mathcal{E}^t) \leftarrow F(\mathbf{X}^t, \mathbf{U}^t, \mathcal{G}^0) \tag{2}$$

$$\Delta \mathbf{v}_{GNN}^t = \mathbf{GNN}^\theta(\mathcal{G}^t) \tag{3}$$

where $F(\cdot)$ is a feature generator function. In parallel, passive forces (cable and gravity) are computed analytically with function $H(\cdot)$ and transformed to node-level velocity changes, $\Delta \mathbf{v}_{PF}^t$, via mapping function $M(\cdot)$.

$$\Delta \mathbf{v}_{PF}^t = M(H(\mathbf{X}^t, \mathbf{U}^t)) \tag{4}$$

Finally, $\Delta \mathbf{v}_{GNN}^t$ and $\Delta \mathbf{v}_{PF}^t$ are integrated up with the semi-implicit Euler scheme to predict next node states $(\mathbf{p}^{t+1}, \mathbf{v}^{t+1})$ and mapped back to rigid-body state $\mathbf{X}^{t+1}$ with $M'(\cdot)$.

$$\mathbf{v}^{t+1} = \mathbf{v}^t + \Delta \mathbf{v}_{PF}^t + \Delta \mathbf{v}_{GNN}^t \tag{5}$$

$$\mathbf{p}^{t+1} = \mathbf{p}^t + \mathbf{v}^{t+1} \Delta t \tag{6}$$

$$\mathbf{X}^{t+1} = M'(\mathbf{p}^{t+1}, \mathbf{v}^{t+1}) \tag{7}$$

This procedure serves as a single simulation step in our framework. Trajectory rollouts can then be generated by applying this process for $K$ number of steps in an auto-regressive manner $(\mathbf{X}^0, \mathbf{X}^1, \ldots, \mathbf{X}^K)$, as shown in Fig. 2. The details of a single step are depicted in Fig. 3.

The **actuation and cable forces** are computed analytically with linear models. For actuation, there are motors that act on cables to change their rest lengths. The cables are modeled as linear springs using Hooke's law, but they only allow tension and not compression. It should be noted that in sim-to-sim experiments, these linear models are exact, but in reality, the cables and the motors can have non-linear components that the GNN can potentially compensate for.

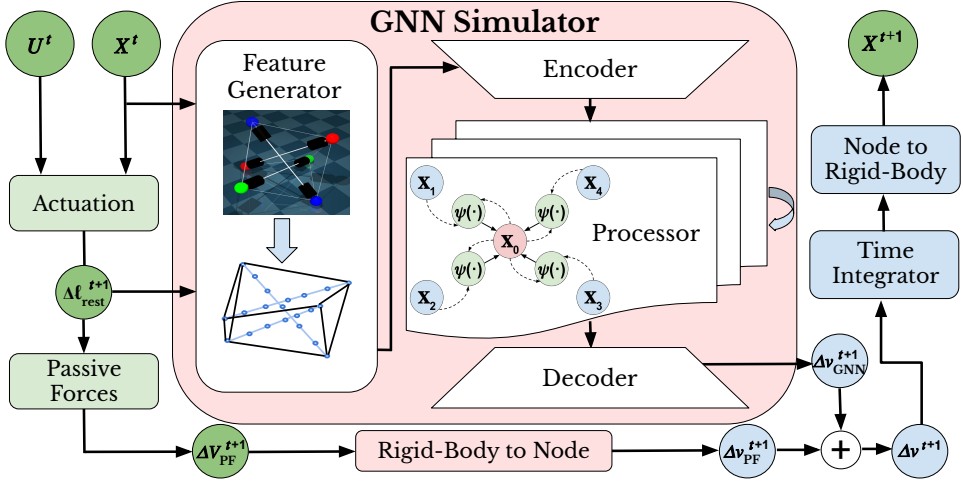

Figure 3: A single simulation step. The current rigid-body state $\mathbf{X}^t$ and controls $\mathbf{U}^t$ are used to analytically compute the cable's rest length changes $\Delta l_{rest}^{t+1}$ and the passive force induced velocity changes $\Delta \mathbf{V}_{PF}^{t+1}$. $\Delta \mathbf{V}_{PF}^{t+1}$ is mapped to node-space velocity changes $\Delta \mathbf{v}_{PF}^{t+1}$. $\mathbf{X}^t$ and $\Delta l_{rest}^{t+1}$ are used to generate node and edge features. They are then encoded, processed through multiple message-passing steps, and decoded to the predicted $\Delta \mathbf{v}_{GNN}^{t+1}$. Finally, $\Delta \mathbf{v}_{PF}^{t+1}$ and $\Delta \mathbf{v}_{GNN}^{t+1}$ are summed, time-integrated, and mapped to the predicted next rigid-body state $\mathbf{X}^{t+1}$.

**Graph structure** The graph $\mathcal{G}$ is composed of body nodes $\mathcal{N} = \{N_i\}$, intra-body edges $\mathcal{E}^{body} = \{E_{ij}^{body}\}$, contact edges $\mathcal{E}^{con} = \{E_j^{con}\}$, and cable edges $\mathcal{E}^{cable} = \{E_{ij}^{cable}\}$. Upon initialization, the topology graph $\mathcal{G}^0$ is generated, containing only the graph structure and node body-frame positions $\{x_i^B\}$. At time step $t$, node and edge feature vectors $(N_i^t, E_{ij}^t)$ and node states in the world frame $\mathbf{x}_i^t = (\mathbf{p}_i^t, \mathbf{v}_i^t)$ are computed based on $(\mathbf{X}^t, \mathbf{U}^t)$ and are added to the topology graph $\mathcal{G}^0$ to form $\mathcal{G}^t$. Each rod is represented by a subgraph consisting of a set of connected body nodes along its center axis formed by decomposing the rod into simpler primitive rigid bodies, as shown in Fig. 4. Rod subgraphs are connected with each other via cable edges at nodes corresponding to the cables between pairs of rod end caps. A special ground body node is

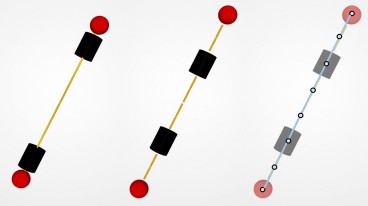

Figure 4: Graph structure for a single rod: (left) original rod; (middle) rod decomposed to multiple primitive bodies; (right) rod nodes and edges.

constructed with dynamically generated contact edges connecting the ground body node to rod body nodes that are within a radius $r_g$. All of these together form a full graph that represents the tensegrity robot. Examples of the graph structures for 3-bar and 6-bar tensegrities can be seen in Fig. 1.

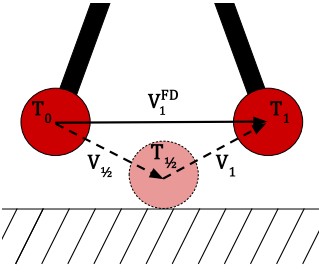

Figure 5: Difference between finite-difference velocity $\mathbf{v}_1^{FD}$ vs. instantaneous velocity $\mathbf{v}_1$ in a single step where a collision occurs.

**Nodes** The set $\mathcal{N}$ of nodes represents primitive rigid bodies that, together, compose the rods. For the GNN to be translation-equivariant, velocities, instead of positions, are used as input features. More detailed descriptions of the features can be found in Appendix A. In addition, the proposed approach only requires the average velocity $\mathbf{v}_i^{FD}$, which can be computed from positional data as $(\mathbf{p}_i^t - \mathbf{p}_i^{t-1})/\Delta t$, as shown in Fig. 5. Previous works in traditional physics simulators [52, 23] that model contact typically compute time-of-impact within a time step and split the step into a pre-contact and post-contact phase for accurate simulation. This computation requires accurate approximations of instantaneous velocities, something hard to infer given only positional data. In contrast, the GNN is able to learn over the average velocities and accurately incorporate contact dynamics in its predictions.

**Edges** There are three types of edges: (i) body edges $E_{ij}^{body}$, (ii) cable edges $E_{ij}^{cable}$, and (iii) contact edges $E_{ij}^{con}$. The body edges $E_{ij}^{body}$ connect nodes belonging to the same rod. These edges learn rigid-body constraints, but do not strictly enforce them. This allows for a small degree of softness that can aid in the learning process. The cable edges $E_{ij}^{cable}$ connect end cap nodes belonging to different rods with physical cable attachments. The contact edges $E_{ij}^{con}$ connect body nodes to the ground node when they are within a user-specified distance. While contact is often viewed as a constraint, this formulation models contact as a unique edge type in the graph neural network due to the fact that contact introduces discontinuous motion with complex friction dynamics.

**GNN architecture - encoder** The encoder is a multi-layer perceptron (MLP), which receives the input feature vectors $N_i$ or $E_{ij}$ and outputs a latent vector embedded in a larger dimensional space. There is a dedicated encoder for each type of node and edge of the graph. In our setup, there is one node encoder, $\mathbf{MLP}_{\mathcal{N}}^{enc}$, and three edge encoders, $\mathbf{MLP}_{\mathcal{E}_{body}}^{enc}$, $\mathbf{MLP}_{\mathcal{E}_{cable}}^{enc}$, $\mathbf{MLP}_{\mathcal{E}_{con}}^{enc}$.

**GNN architecture - processor** The processor is a sequence of message-passing steps that aims to learn the latent dynamics. At each message-passing step $l$, all latent edge vectors are updated from the current latent edge vectors and the two connecting latent nodes' vectors. Then, node vectors are updated from aggregating new edges and passed through another MLP. This process is repeated $L$ times with $L$ different MLPs that do not share weights.

$$E_{ij}^l = \mathbf{MLP}_l^{MP}(N_i^{l-1}, N_j^{l-1}, E_{ij}^{l-1}) \tag{8}$$

$$N_i^l = \mathbf{MLP}_l^{update}(N_i^{l-1}, \sum E_{ij}^{body,l}, \sum E_{ij}^{cable,l}, \sum E_{ij}^{con,l}) \tag{9}$$

**GNN architecture - decoder** The decoder is also an MLP that takes the last latent node vectors and outputs the predicted velocity changes $\Delta v_i^t$ per node, which are then used to integrate up to velocities and positions. $\Delta v_i^t$ are predicted instead of position or velocities so that the GNN's predictions would not violate the underlying governing differential equations.

$$\Delta \mathbf{v}_{GNN,i}^t = \mathbf{MLP}^{dec}(N_i^L) \tag{10}$$

**Loss** The loss function is the mean squared error (MSE) per node between the predicted and (computed) ground truth (GT) velocity changes.

$$\Delta \mathbf{v}_{GT}^{t+1} = ((\mathbf{p}_{GT}^{t+1} - \mathbf{p}^t)/\Delta t) - \mathbf{v}^t \tag{11}$$

$$\mathcal{L}(\mathcal{G}^t, \Delta \mathbf{v}_{GT}^{t+1}) = \frac{1}{B} \sum (\mathbf{GNN}_\theta(\mathcal{G}_i^t) - \Delta \mathbf{v}_{GT,i}^{t+1})^2 \tag{12}$$

As the experiments show, the loss must be defined at the node level instead of the rigid-body state level. A loss at the state level does not drive the GNN to learn node dynamics since the node output is averaged when mapping back to the rigid-body state. In addition, $\Delta \mathbf{v}_i$ are used instead of positions or velocities as the acceleration errors are not functions of the time step. As time step decreases (often needed for accuracy and stability), the positional error magnitude becomes smaller, causing the update gradients to vanish.

## 4  Experimental Results

The approach is compared against the previous first-principles differentiable engine (DPE) in both simulation and reality. There is also an ablation on which parts of the engine should be computed with first-principles and which should be learned via a GNN, as well as which quantities to compute the loss over. Lastly, the proposed approach is compared against the surface-based (mesh and meshless) GNN representations from the literature on learning rigid-body simulators.

**Evaluation metrics:** The metrics are the average positional, rotational, and ground-penetration error over a trajectory rollout $(\mathbf{X}^0, \mathbf{X}^1, ..., \mathbf{X}^K)$. The positional error $\mathbf{e_{pos}}$ is measured as the absolute distance between the predicted and GT robot center of mass $\mathbf{P}$ normalized by the length of a rod $L_{rod}$. The rotational error $\mathbf{e_{rot}}$ is measured as the angular difference between the rods' predicted and GT center axes $\hat{\mathbf{r}}$. The ground penetration error $\mathbf{e_{pen}}$ is the absolute penetration distance into the ground *beyond* what is seen in the GT data and normalized by the length of the rod, where $z$ is the height of the point closest to the ground on the robot:

$$\mathbf{e_{pos}} = \frac{1}{L_{rod}K} \sum |\mathbf{P}^i_{GT} - \mathbf{P}^i_{pred}|, \quad \mathbf{e_{rot}} = \frac{1}{K} \sum \cos^{-1}(\hat{\mathbf{r}}^i_{GT} \cdot \hat{\mathbf{r}}^i_{pred})$$

$$\mathbf{e_{pen}} = \frac{1}{L_{rod}K} \sum \max(\min(z^i_{GT}, 0) - \min(z^i_{pred}, 0), 0)$$

**Sim-to-sim evaluation** Two types of GT trajectories are generated in MuJoCo: (1) active trajectories with control signals, and (2) passive trajectories where the tensegrity robot is initialized with random height and linear velocity. This is done for both a 3-bar and a 6-bar tensegrity robot. Forty-six trajectories were generated, amounting to $\sim$ 17 minutes of data. These data were split into 50% training, 25% validation, and 25% testing by trajectory. The models were trained for a fixed number of epochs, and the model with the lowest validation loss was saved and executed on the held-out test set. The baselines are the DPE, an MLP trained to predict rod dynamics, and an MLP trained to predict all rods simultaneously.

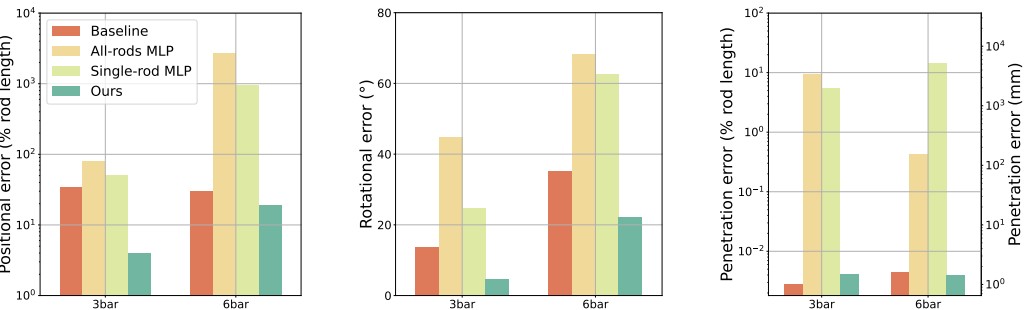

Figure 6: Sim-to-sim results for the 3-bar and 6-bar robots. The approach is compared against 3 alternatives: the baseline differentiable engine, an MLP for each rod, and an MLP for all rods.

In simulation where we have access to high-frequency, full state information, the proposed method strongly outperforms the baselines as highlighted in Fig. 6. It improved upon the differentiable physics engine by 30% and 9° for the 3-bar tensegrity and 11% and 13° for the 6-bar tensegrity. The 6-bar tensegrity was more difficult to learn over due to increased complexity. The 6-bar robot has a smaller base of support than the width of the robot, which causes it to be unstable with small momentum. For the 3-bar tensegrity, the base's width is equal to the robot's width, so it is more stable. The MLP variants performed poorly, especially with high penetration errors, reinforcing the results seen in previous works showing vanilla neural networks have difficulty learning discontinuous contact dynamics. Although the penetration error is higher in the proposed method than the baseline, both the proposed model and baseline have low penetration errors, i.e., below 0.01% (0.036mm) in simulation results. Furthermore, the authors hypothesize that, with the graph formulation, the no-penetration constraint is "softened," and hence the contact dynamics are smoothed and learning is eased.

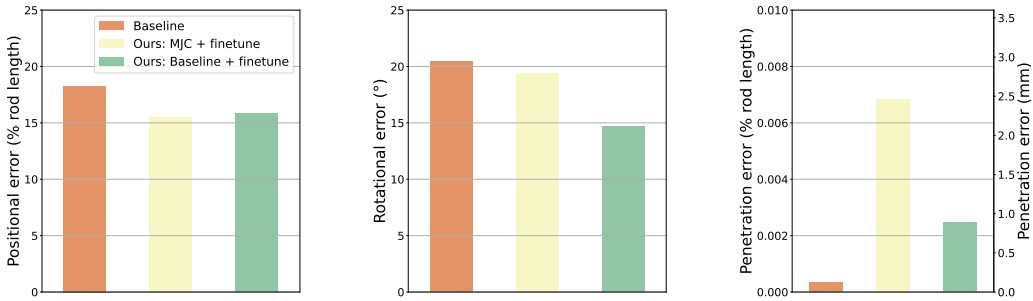

Figure 7: Physical 3-bar tensegrity experimental results. Compared our method with warm starting with either MuJoCo simulator data or DPE simulator data.

**Real 3-bar tensegrity** For the real 3-bar tensegrity robot, 16 trajectories were collected of $\sim$ 1.5 minutes each that include straight rolling, clockwise and counterclockwise rolling, crawling, and random cable actuation motion. This dataset is split to eight trajectories for training, four for validation, and four for testing. The collected data only track the center of the end caps' positions. This means that the rotational component about the rod center axes and the instantaneous velocities

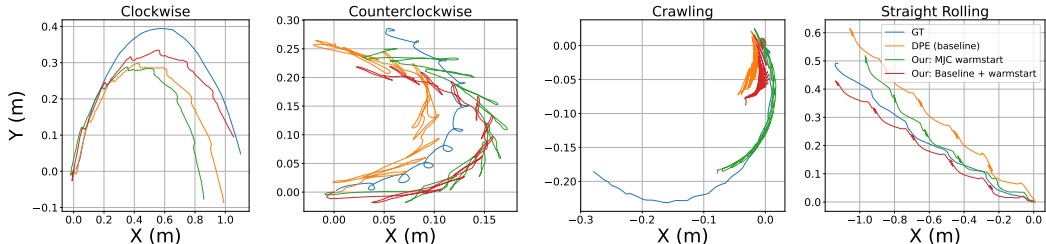

Figure 8: Plots of tensegrity robot's center-of-mass starting from the origin, (0, 0). Trajectories range from 45 to 75 seconds in length with a stepsize of 10ms. Four gaits in the test set comparing the baseline (DPE) and the learned simulators (with MJC and baseline warmstarts) against ground truth. Rendering of these trajectories can be seen in the attached video.

are unavailable. Furthermore, the data are captured at a lower frequency with non-uniform step sizes, including control signals. Due to these issues, additional steps to the current training and testing process are used to work with real data. A PID controller is included in the loop reflecting the operation of a PID controller on the real platform that receives cable length targets and generates the low-level controls. This introduces temporal differences, and the evaluation criteria focus on measuring errors at the last frame per gait cycle. Finally, the engine is pre-trained over data generated from a simulator (MuJoCo and baseline), then fine-tuned over the entire trajectories of the real robot.

The GNN model can still learn over observations of a real tensegrity and improve upon it, as seen in Fig. 7. The improvements are more incremental relative to the full observability results with about 3% improvement over the baseline in positional error and about 1°and 5°over the baseline in rotational error. A potential reason is that fine-tuning over long trajectories may not be ideal. The long trajectories produce deep computational graphs, which can cause vanishing gradients. Secondly, the long trajectories also have noisy observations that produce conflicting gradients when trained in a single step. Lastly, the PID controller also influences the number of steps and input control signals, changing the internal structure of the computational graph and, subsequently, the loss function that the model is optimizing for. Thus, it is not necessarily the case that the gradients are still pointing in a downward direction for the changed loss function. Qualitatively, the GNN models trained in this fashion are still able to exhibit improved predictive capabilities in terms of the test trajectories of Fig. 8. Similar to the simulation results, even though the GNN results have higher penetration errors than the baseline results, the GNN results are still below 0.008% (0.03mm).

| Model | Positional Error (%) | Rotational Error (°) | Penetration Error (% / mm) |
|---|---|---|---|
| All analytical (baseline) | 34.30 | 13.74 | 0.0028 / 0.01 |
| All analytical + GNN residual | 8.87 | 8.93 | **0.0017 / 0.006** |
| Analytical passive forces + GNN contact (ours) | **4.03** | **4.69** | 0.0041 / 0.015 |
| GNN passive forces + GNN contact | 9.35 | 10.97 | 0.0032 / 0.01 |

Table 1: Ablation study that compares GNN in different parts of the simulation workflow.

**Ablations** Table 1 evaluates (i) an all analytical model (DPE baseline), (ii) analytical with a GNN global residual, (iii) analytical passive forces (cable and gravity) computation and the GNN for contact (our chosen method), and (iv) a single GNN that learns both passive forces and contact. The data used here are the 3-bar tensegrity data from MuJoCo also used for the above sim-to-sim experiments. All GNN variants outperformed the all analytical baseline, and the proposed hybrid approach showed the best results in positional and rotational error.

Table 2 evaluates the choice of the loss function. Two alternatives to the proposed approach are considered that operate in rigid-body space: (i) a positional loss over the rod end cap positions, which combines positional and rotational errors into one metric; (ii) a 6D pose loss comprising a Euclidean distance between centers of mass and an angular difference between center axes per rod. The end points' positional loss outperformed the 6D pose loss, but the proposed node-based loss strongly outperformed both rigid body-based losses as it better enforces node dynamics.

| Model | Positional Error (%) | Rotational Error (°) | Penetration Error (% / mm) |
|---|---|---|---|
| Position + orientation loss | 126.64 | 32.95 | 0.0086 / 0.031 |
| End point positions loss | 69.07 | 28.24 | 0.0050 / 0.018 |
| Node position loss (ours) | **4.03** | **4.69** | **0.0041 / 0.015** |

Table 2: Ablation study that compares loss taken in rigid-body space and node space.

**Mesh/Surface representations vs. multi-object representations** Table 3 compares the proposed approach against alternative choices for the graphical representation for the GNN previously used for general rigid-body GNN simulators i.e., the general mesh [9, 10], and surface-based [11] representations. The table reports training wall-clock time, inference wall-clock time on the GPU, and inference wall-clock time on a single CPU.

| Model | Pos Error | Rot Error | # of Nodes | # of Edges | GPU Train (hours) | CPU Train (hours) | GPU Inference (s/step) | 1 CPU Inference (s/step) |
|---|---|---|---|---|---|---|---|---|
| Mesh | 38.41% | 67.51° | 603 | 7632 | 58.66 | n/a | 0.0228 | 0.1238 |
| Surface | 49.60% | 90.53° | 600 | 1212 | 25.51 | n/a | **0.0171** | 0.0207 |
| Object (ours) | **8.78%** | **8.84°** | **22** | **60** | 7.69 | 12.56 | 0.0237 | **0.0153** |

Table 3: Performance metrics for different graphical representations for the GNN.

The proposed object-based representation is much more accurate as well as efficient. The mesh and surface-based representations did not converge to a good rollout accuracy for the same epochs. They also need more time, model size, and node density to achieve reasonable accuracy, all of which increases the computational resources required. The proposed approach can be trained via CPU only with a small increase in training time. For inference over GPU, the proposed representation is mildly slower than the mesh and surface ones. When inference is performed over a single CPU, the proposed object-based representation is faster than the alternatives.

## 5 Limitations

While the proposed engine outperforms the alternatives in data captured from a real 3-bar tensegrity, it performs more effectively on data captured from simulation with full observability. This is due to several key differences between the two setups: (i) real sensing data only provide the center position of each end cap, providing only five out of the six DoFs of each rod (missing the orientation about its center axis); (ii) real state estimation is not directly estimating instantaneous linear and angular velocities; (iii) real observations have lower frequency than what is possible in simulation and may be available at non-uniform time intervals; (iv) the real PID controller was a black box that could not be perfectly modeled in simulation. A potential direction to overcome these partial observability limitations corresponds to applying trajectory smoothing and using additional sensing information to estimate each rod's orientation about its axis. Smoothing can help identify the set of states that best explain the entire observed trajectory so that it respects physical constraints.

Another limitation is that the proposed method only considers flat ground, as a step in modeling contact dynamics for tensegrity structures using GNNs. To deal with non-flat terrains and obstacles, the plan is to generalize the distance-to-ground feature to a signed-distance field given the terrain. This would be a high-dimensional feature, so the encoding of this information would need to be explored. Additionally, data diversity would increase, thus increasing data requirements.

## 6 Discussion

This work shows that representing tensegrity robots as object-based graphs allows GNNs to learn complex contact dynamics and improves simulation accuracy for differentiable engines. It also provides computational benefits over alternatives. This observation can have broader implications for modeling robotic platforms via GNNs, especially those that are difficult to model analytically and which exhibit a graphical structure. A potential target is adaptive hands [53] that are also cable-driven, have compliant joints, and can be graphically represented. The simulator can also be used for learning controllers that achieve more sophisticated gaits and skills.

**Acknowledgments**

N. C. and M. A. were supported in part by the National Science Foundation (NSF) under awards CCF-2110861, IIS-2132972, IIS-2238955, and CCF-2312220 as well as a research gift from Red Hat, Inc. In addition, N. C. and K. B. were supported in part by the NSF under award IIS-1956027. W. R. J. and R. K. B. were supported by the NSF under grant no. IIS-1955225. Any opinions, findings and conclusions, or recommendations expressed in this material are those of the authors and do not necessarily reflect the views of the NSF.

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

# Supplementary Material

## A    Graph features

**Body nodes** Each node has the features $N_i = \{m_i^{-1}, I_i^{-1}, \mathbf{v}_i^{FD}, d_g\}$, where $m_i^{-1}$ is the reciprocal of the mass, $I_i^{-1}$ is the inverse of the inertia tensor in the principal directions, $\mathbf{v}_i^{FD}$ is the first-order finite-difference velocities defined as $(\mathbf{p}_i^t - \mathbf{p}_i^{t-1})/\Delta t$, and, lastly, $d_g$ is the distance from the ground. The finite-difference velocities are not instantaneous as in traditional physics simulators. They are instead average velocities over the time step. Previous work on differentiable engines [52] computes time of impact within a time step for accurate simulation. The proposed approach only considers average velocity instead, without splitting the time step, as shown in Fig. 5. The GNN is able to learn over average velocities and accurately incorporate the contact dynamics in its predictions.

**Body edges** $E_{ij}^{body}$ have features $E_{ij}^{body} = \{\mathbf{d}_{ij}, \|\mathbf{d}_{ij}\|, \mathbf{d}_{ij}^U, \|\mathbf{d}_{ij}^U\|\}$, where $\mathbf{d}_{ij}$ is the displacement vector between two body nodes and $\mathbf{d}_{ij}^U$ is the displacement vector between two body nodes in the undeformed, body-frame state.

**Cable edges** $E_{ij}^{cable}$ have features $E_{ij}^{cable} = \{\mathbf{d}_{ij}, \|\mathbf{d}_{ij}\|, \hat{\mathbf{d}}_{ij}, \mathbf{v}_{ij}^{rel}, L_{ij}^{rest}, K_{ij}, c_{ij}\}$, where $(\mathbf{d}_{ij}, \mathbf{v}_{ij}^{rel})$ are the displacement and relative velocity vectors between end cap nodes $i$ and $j$, and $(L_{ij}^{rest}, K_{ij}, c_{ij})$ are the cable's rest length, stiffness, and damping, respectively.

**Contact edges** $E_{ij}^{con}$ have features $E_{ij}^{con} = \{d_g, \hat{\mathbf{n}}, v_{rel}^{norm}, v_{rel}^{tan}\}$, where $d_g$ is the minimum signed-distance to the ground, $\hat{\mathbf{n}}$ is the contact normal unit vector, and $v_{rel}^{norm}$ and $v_{rel}^{tan}$ are the normal and tangential components of the relative velocity.

## B    Training details

**Hardware and implementation** All methods are trained and tested on a machine equipped with a Nvidia RTX 4090 GPU and an AMD Ryzen 7950 16-core, 4.5 GHz CPU. All simulators were built and executed using PyTorch and PyGeometric.

**Training strategy** For models trained with simulation data, a curriculum learning strategy is employed where the trajectory rollout length is progressively increased. Initially, the model is trained to predict the immediate 1-step ahead state for 200 epochs. The model train loss and validation loss will start to flatten out and further training tends to see an increase in full trajectory error. Next, the model is trained to perform a 2-steps look ahead for 100 epochs, 4-steps ahead for 50 epochs, and finally, 8-steps ahead for 25 epochs. It is observed that further increase in rollout length does not decrease error. This procedure allows the model to incrementally experience the errors it makes during rollout and to adjust to them.

For the mesh and surface-based models, the models were trained based on the procedures described in prior work [9, 10, 11]. These models were only trained with 1-step roll-out lengths and with Gaussian random-walk noise of $5 \times 10^{-4}$ to node positions upon input to the model, for 800 epochs.

**Training hyperparameters** The models were trained with progressively decreasing learning rates of $10^{-5}, 10^{-6}, 10^{-7}$, and $10^{-8}$. These learning rates correspond to the n-steps look ahead curriculum phases as detailed above. A mini-batch size of 128 was used during training. The Adam optimizer with a weight decay of $10^{-2}$ was used.

**Data Augmentation** At the start of each training step, the mini-batch is rotated by a random angle $[-\pi, \pi]$ about the z-axis.

## C Model details

**Network architecture** All the multi-layer perceptron (**MLP**) models have two layers, ReLU activations, and residual connections. All **MLP**s, except the decoder, apply LayerNorm on their outputs. For the surface-based vs multi-object experiments, the width and latent vector dimension sizes were 64. For all other experiments, the dimension sizes were 128. Lastly, models for the 3-bar tensegrity had four message-passing steps while the models for the 6-bar tensegrity had 10 message-passing steps due to the larger graph. All inputs were normalized to zero-mean and unit variance.

**Rigid-body state to node states** Let $\mathcal{T}^t$ be the rigid-body transformation that takes the rigid body from its body frame to the world frame at time step $t$. To compute the node states, $\mathbf{p}_i^t, \mathbf{v}_i^t$, the same transformation is applied to the node positions in the body frame:

$$[\mathbf{p}_i^t, \mathbf{v}_i^t] = [\mathcal{T}^t(\mathbf{p}_i^B), (\mathcal{T}^t - \mathcal{T}^{t-1})(\mathbf{p}_i^B)/\Delta t]$$

**Node states to rigid-body state** Since the node positions are not strictly enforced to maintain relative distance from one another, there can be shape violations. Hence, an approximation of the rod's pose is needed. $\mathbf{P}^t$ is approximated as the average of the node positions. $\mathbf{R}^t$ is computed as the minimal rotation needed to rotate the z-axis unit vector to rod's center axes $\hat{\mathbf{r}}$, approximated with the unit vector pointing from one rod endpoint to the other. A more principled method, such as shape matching [54], can be used if the graph is more complex. Subsequently, the rigid-body velocities are the first-order approximation of velocities needed to take $[\mathbf{P}^t, \mathbf{R}^t]$ to $[\mathbf{P}^{t+1}, \mathbf{R}^{t+1}]$.

**Cable model** The cable model is a linear model following Hooke's law that allows for tension but not compression. It computes the cable force $\mathbf{F}_{cable}^t$ at time $t$ based on the stiffness component $F_K^t$ and the damping component $F_c^t$:

$$F_K^t = \begin{cases} K(l_{rest}^t - \Delta x^t) & \text{if } l_{rest}^t \geq \Delta x^t \\ 0 & \text{otherwise} \end{cases}$$
$$F_c^t = c(\mathbf{v}_{rel}^t \cdot \hat{\mathbf{x}}^t)$$
$$\mathbf{F}_{cable}^t = (F_k^t - F_c^t)\hat{\mathbf{x}}^t$$

where $K$ is the cable stiffness, $c$ is the cable damping, $l_{rest}^t$ is the rest length at time $t$, $\Delta x^t$ is the distance between the cable attachment points at time $t$, $\mathbf{v}_{rel}^t$ is the relative velocity between the cable attachment points, and $\hat{\mathbf{x}}^t$ is the unit direction pointing from one attachment point to the other.

**Actuation model** The actuation model consists of a linear model of a motor that receives an input control signal $u^t$ between $[-1, 1]$ and acts on a cable by changing the cable's rest length $l_{rest}^t$ at time t:

$$\omega^t = s\omega_{max}u^t$$
$$\Delta l_{rest} = 0.5(\omega^t + \omega^{t-1})r_{winch}\Delta t$$
$$l_{rest}^t = l_{rest}^{t-1} - \Delta l_{rest}$$

where $\omega^t$ is the angular velocity of the motor at time $t$, $s$ is an input speed parameter $[0, 1]$, $\omega_{max}$ is the maximum angular velocity the motor can achieve, $\Delta l_{rest}$ is the change in rest length due to the motor, $r_{winch}$ is the winch radius of the motor, and $\Delta t$ is the time step size.

## D Tensegrity Robot Details

Table 4 provides the values for physical parameters of the real tensegrity robot. These measurements were used in order to set the corresponding parameters in simulation to the same value. The parameters are not learned by the differentiable physics engine.

| Attribute | Measurement |
|---|---|
| inner rod length | $0.325m$ |
| inner rod radius | $0.0016m$ |
| inner rod mass | $3.8g$ |
| end cap radius | $0.0175m$ |
| end cap mass | $10.5g$ |
| motor radius | $0.0175m$ |
| motor length | $0.045m$ |
| motor offset (center to center) | $0.1175m$ |
| motor mass | $35.3g$ |
| short cable stiffness | $10^5 N/kg$ |
| short cable damping ratio | $10^3 N \cdot s/m$ |
| long cable stiffness | $10^4 N/kg$ |
| long cable damping ratio | $10^3 N \cdot s/m$ |

Table 4: Physical tensegrity robot measurements used as simulation parameters

# E  MuJoCo Setup

MuJoCo was the simulator chosen to be the ground truth data in the sim-to-sim experiments. The data were generated using the following setup:

- Rigid bodies and cables were set with the same mass, geometric properties, and stiffnesses as the real robot shown in Table 4 of the appendix;

- Friction coefficients were set to 0.5 for tangential, 0.005 for torsional, and 0.9 for rolling.

- The time step size was 10 ms, and a semi-explicit Euler time integration scheme was employed.

- The tendon model was adopted but modified to only apply tension but no compression forces.

- The authors executed the rolling and crawling controls developed for the real robot, and a simulated PID controller, to generate trajectories as the training, validation, and testing data sets.

- Additionally, the authors generated drop and throwing trajectories by initializing robots at random heights and random linear velocities.

# F  Multi-Layered Perceptron Baselines

**MLPs** were used in the sim-to-sim experiments to show why a specialized architecture based on GNNs is beneficial. In order to best compare against the **MLP** baselines to the GNN, the authors tried to keep the neural network size the same between the two. This resulted in the MLP comparison points having:

- Layer widths of 128
- (2 + (number of message passes in GNN * 4) + 2) number of layers
- Residual connections every 2 layers
- LayerNorms every 2 layers, except at the very last layer
- ReLU activation functions
- Trained with the same number of epochs ($\sim 375$) as the GNN

# G  Real Robot Training Procedure

For the real 3-bar tensegrity robot experiments, the following steps were taken:

1. A simulator (DPE or MuJoCo) was tuned to best match the real training set
   - DPE learned via gradient descent;
   - MuJoCo contact parameters were searched via random search

2. Dense, high-frequency, and fully observable data matching training trajectories were generated in simulation

3. The proposed GNN model was then pretrained over the generated simulation training data using the training setup and strategy descibed in Appendix B.

4. Then, the pretrained GNN model was fine-tuned by training over the real training trajectories, where a single forward pass is a full rollout. This model was trained over 10 epochs at a relatively small learning rate of $10^{-8}$. Additionally, a PID controller was included in the training loop.

