# OpenReview forum: "Learning Differentiable Tensegrity Dynamics using Graph Neural Networks"
_robot-learning.org/CoRL/2024/Conference — CoRL 2024_

### Official Review · Reviewer_uba8 · 2024-07-18
**Interesting paper with nice simulation results**

**Originality:** 3
**Technical Quality:** 3
**Clarity Of Presentation:** 4
**Potential Impact:** 3
**Recommendation:** 3
**Confidence:** 4

**Review:**

Using the graph-like structure of multi-body dynamics in neural network regression is sensible. Investigating this topic from the perspective of tensegrity robots is also quite interesting. On one hand, these robots have a relatively "simple" mechanical design that one can intuitively grasp. On the other hand, the kinematic closed-loop structure is by no means straightforward to model accurately through analytical methods. In particular, when contact forces and actuation are being involved.

Strength:
- Video is top notch.
- GNN model combining robot kinematics with NN regression.
- Paper is well written.
- Hardware experiments.

Weaknesses:
- **C1: Residual learning on velocity level:**
  The author predict the change in velocity using the analytical model and then learn the error in these velocities with a GNN. This is clearly the first thing one would try when doing residual learning on such a mechanical system. However, I have the superstition that most of the errors in your model actual arise from an inaccurate depiction of the forces being involved. It might be easier to learn errors in the impulse/force space than in velocity-space as the latter arises from nonlinear transformations involving the impulses, masses, and system kinematics. In other words, what the authors do is to use the structural knowledge from the robot's kinematics (interaction between body points and constraints) to build a GNN, but there is a lot of structural knowledge in the system's dynamics (interacting between mass and force) that one could use to setup a GNN.

- **C2: Body nodes are not kept at a fixed distance from each other:**
  The authors seem to not enforce explicitly that the model's predictions respect the kinematic constraints (fixed rod length). This can be rather problematic as the constrained system dynamics form a low-dimensional manifold inside the chosen high-dimensional state space. Prediction errors of long feed-forward predictions push your system state of this manifold. In turn, the network predictions deteriorate as it has not seen these "off-manifold" points during training. You could add a regularization term to the loss that punishes kinematic constraint violations or, alternatively, add to your network an implicit function layer that enforces constraint satisfaction (slows down training a lot).

- **C3: Only incremental improvements on real-world data:**
  On the real-world system the model only shows a neglectable 3% improvement. However, learning contact dynamics alongside a quite complicated dynamical system is a difficult task that roboticists are fencing with since years. As the contribution of this work is not on contact modeling, but on exploiting the graph structure in tensigrity robots, I don't mind the model's performance on real-world data. I have the following suggestions for improving the performance of follow-up models: 1) Improve the state observer or feed a history of states to the network such that it can learn to filter the noise in the model inputs. If you unroll a dynamics model over several steps, small input noise can cause large overall prediction errors. 2) Train your model first on short trajectories and then increase the trajectory length after some episodes. This approach benefits model convergence, see e.g. https://docs.kidger.site/diffrax/examples/neural_ode/ 3) Consider also an error in the robots contact point, as the time of impact has a tremendous effect on the simulation.


Minor comments:
- Please share your code and data.
- I assume the authors represent rotation via quaternions. If this is the case, I suggest that you use instead vectorized rotation matrices in your input and predict changes in rotation via quaternions. See e.g. "Learning with 3D rotations, a hitchhiker's guide to SO(3)" Geist et al. and "Deep Regression on Manifolds: A 3D Rotation Case Study" by Bregier.
- Figure 2: The background is distracting.
- Many illustrations use the png format, please turn them into vector graphics. If the figures are not images nor contain thousands of points, there is no reason why you shouldn't.

**Quality Of The Limitations Section:**

2

**Questions For Rebuttal:**

- **Question - C1:** Do the authors feed to the GNN the position of each node? If yes, how do you ensure that the GNNs prediction stays invariant to translation in space?
- **Question 2:** How do you tune the baseline model? Can you share the hyper-parameter search method and its results?
- **Question 3 - C2:**  What is Fig. 8 showing? Is this simply a model rollout? If yes, for how long and what prediction FPS? If the roll-out is longer than 2 seconds, how do you ensure that the predicted state is not diverging?

**Robotics Focus:**

4

**Summary Of Paper:**

The paper proposes a graph neural network architecture to learn the discrete-time forward dynamics of tensegrity robots. Each joint of the robot is modeled as node in a Graph while linear rigid bodies are modeled as a sequence of nodes that are connected by edges. Contacts are modeled as edges between nodes and the ground which is assumed to be flat. The GNN is trained via supervised regression to predict changes in the node states. The utility of the proposed model is being demonstrated in simulation using a Mujoco model as baseline and hard-ware experiments.

**Summary Of Recommendation:**

Nice work that shows how to use structure from robot kinematics to predict dynamics using GNNs. The authors could discuss a bit more critically what design decisions they made to get the model and why. Also the experimental discussion requires some polishing. Albeit, I enjoyed reading this paper.

---

### Official Review · Reviewer_mw3p · 2024-07-19
**Review on Learning Differentiable Tensegrity Dynamics using Graph Neural Networks**

**Originality:** 3
**Technical Quality:** 2
**Clarity Of Presentation:** 4
**Potential Impact:** 2
**Recommendation:** 3
**Confidence:** 4

**Review:**

# Quality
The GNN based model proposed in this work has been validated in both sim-tosim and real experiments, and it has been demonstrated that this method has improved the dynamic modeling quality compared to previous work.
# Clarity
This paper clearly describes the design principles of the GNN based model used, as well as the results of sim-to-sim experiment and real experiment. However, it still lacks some detailed introductions, such as the parameters of GNN.
# Originality
This paper applies GNN to tensegrity robots, which is innovative compared to the original work.
# Strengths
1. This paper demonstrates better dynamic modeling performance for tensegrity robots compared to previous work.
# Weaknesses
1. In sim-to-sim experiments, 3-bar and 6-bar robots are tested and performed well; while in real experiment, only 3-bar robot was tested. It's more convincing to validate the model on real 6-bar robot.
2. The positional and rotational error of the proposed model is much lower than the DPE, but the penetration error is higher, of which the influence is not introduced in the paper.
3. There lacks some detailed introductions on MLP and GNN, such as layers, training epoches.

**Quality Of The Limitations Section:**

2

**Questions For Rebuttal:**

1. Please add more details on the setup of the simulation in MuJoCo.
2. Please point out what DPE is used for comparison, and is it state-of-the-art?
3. Please introduce the details on MLP, such as layer numbers, epoches, and can it be improved by adding more layers or other ways?
4. Please add more details on the finetune during the real 3-bar tensegrity robot experiment.

**Robotics Focus:**

4

**Summary Of Paper:**

This work proposes a learning method using graph neural networks (GNNs) to improve tensegrity robots modeling. The graph representation leverages the natural-like cable connectivity between the rod end caps. The model is used for 3-bar and 6-bar tensegrity robots in simulation and 3-bar one in reality. The result shows higher accuracy and efficiency than previous work. This work shows a good application of GNN for modeling the dynamics of a hybrid rigid-soft mobile robot.

**Summary Of Recommendation:**

This paper shows the potential advantage of graph neural network (GNN) for applicaitons on hybrid rigid-soft robot, which inspires readers to model the dynamics to robot using GNN.

---

### Official Review · Reviewer_z5sV · 2024-07-31
**Learning Differentiable Tensegrity Dynamics using Graph Neural Networks**

**Originality:** 3
**Technical Quality:** 4
**Clarity Of Presentation:** 3
**Potential Impact:** 3
**Recommendation:** 3
**Confidence:** 4

**Review:**

Strength
+ Innovatively applying GNNs to model the complex dynamics of tensegrity robots
+ Providing a detailed description of the proposed method
+ Conducting thorough experiments, comparing this approach with previous differentiable engines and GNN-based simulators.

Weaknesses

- The model assumes a flat ground, which may not generalize well to varied terrains encountered in practical scenarios. The authors acknowledge this limitation and suggest the need for more sophisticated models.

- The ablation studies provided are informative but somewhat limited.  Further exploration into different GNN architectures or alternative representations for the tensegrity structures could provide deeper insights.

- The discussion on handling noisy real-world data and the effect of imperfect PID control loops is brief. More detailed analysis or strategies to mitigate these issues would be valuable, especially since real-world data is often imperfect.

- It appears that the template has been manipulated, potentially to fit more content into the document. This can negatively impact readability.

**Quality Of The Limitations Section:**

2

**Questions For Rebuttal:**

- Why was the specific GNN architecture chosen, and how does it compare to other possible architectures, such as those incorporating attention mechanisms or recurrent elements? Would alternative architectures potentially improve the model's performance, especially in handling temporal dynamics and varying contact conditions?

- Given the assumption of flat ground in the current model, what modifications would be necessary to accommodate complex terrains or non-flat surfaces? Can you discuss the potential challenges and solutions for expanding the model to more diverse environments?

- How does the model scale with increasing complexity in tensegrity structures or more extensive simulation environments?

**Robotics Focus:**

4

**Summary Of Paper:**

The paper introduces an approach to modeling the dynamics of tensegrity robots using Graph Neural Networks (GNNs). The paper's primary strength lies in its innovative application of GNNs to model the complex dynamics of tensegrity robots. In particular, they propose a framework that represents tensegrity robots as graphs, utilizing the graph-like structure formed by the robots' cables and rods. The approach aims to overcome challenges in modeling and controlling tensegrity robots due to their complex dynamics and high degrees of freedom. The method leverages GNNs to predict the contact dynamics and improves upon previous differentiable physics engines. The results show better accuracy and computational efficiency when compared to existing methods, demonstrating the potential of GNNs in this domain.

**Summary Of Recommendation:**

The paper presents an advancement in the modeling of tensegrity robot dynamics using GNNs.

---

### Official Review · Reviewer_Luga · 2024-08-01
**Interesting application of graph-neural-networks to tensegrity structures.**

**Originality:** 2
**Technical Quality:** 3
**Clarity Of Presentation:** 3
**Potential Impact:** 2
**Recommendation:** 3
**Confidence:** 4

**Review:**

# Soundness of claims, significance of work, novelty of contribution, and relevance to CoRL

The authors’ provide a technique for modeling a tensegrity structure and a process for training a GNN to match experimental observations. Furthermore, the authors’ claim the method is more computationally efficient (faster) and more accurate in predicting the behavior of tensegrity structures.

The technique has immediate application in modeling “semi-soft” structures and possibly mechanisms that have numerous active tensile elements, such as tendon-driven robotics and musculoskeletal modeling.

The approach is novel and important beyond previous work using rigid-body simulators and differentiable physics, because it provides a very direct data-driven approach for fitting the simulation to observed phenomena.

The claims on performance are weak, since this paper does not provide significant comparison with previous methods. Furthermore the comparisons that are provided show comparable or modest improvements over the next-best method. Suggestion: soften the statements on computational speed and clarify what makes this method more accurate OR provide further evidence that this method is superior against a wider range of simulators referenced.

# Weaknesses and limitations

The paper proposes an interesting methodology that happens to associate the physical embodiment of a tensegrity structure with the graph structure of GNNs. There is a certain poetry in this composition of “graph-like-things”, but it isn’t obvious why only the contact portion of the problem was modeled with a data-centric approach. Suggestion: Add some framing (or highlight if I missed it) why (a) a hybrid approach was taken (b) why only contact was modeled with the GNN.

The approach as laid out in Section 3 is reasonably clear, but is abstracted enough from the actual physics to beg a few questions. What is the physical intuition on “average velocity”? This reviewer would expect some justification via impulse-and-momentum or at least some excuse as to why this doesn’t violate causality. Suggestion: Either in prose (even better if done with some equations) connect the GNN model to how this directly relates to the rigid-body simulator and explain why this should converge to Newton's laws.

The approach appears to be “position-based”, which is presumably a very “stiff” problem in the context of contact. This appears to be the case given the amount of penetration error observed in the experiments. Suggestion: In the discussion explain why this method is either superior to other approaches or how this might be mitigated (e.g. would formulating this in terms of impulses result in better/worse performance?).

# Correctness

This author would prefer more explicit descriptions of the underlying physics being modeled; however, what is presented appears to be consistent with no visible errors.

# Clarity

The paper is reasonably clear. The exact layout of what pieces are used in the full simulation vs training and ground-truth is a little challenging to follow. Suggestion: A similar figure to Fig 3 describing the ground-truth simulation in context to the training would be very helpful.

The various error-plots use normalized units. I am sure this is very helpful, but it begs questions from a reader of the paper about the absolute values. It would be helpful to have an appendix or another set of plots that demonstrated the penetration in meters, for example.

This author may be an outlier, but I did expect to see some explicit connection to newton’s laws of motion. It is very difficult to evaluate if this method is destined to massive overfitting, because it is just a high-parameter curve-fit or if there are obvious advantages that could lead to guarantees on physical correctness (conservation of momentum, passivity, etc). Suggestion: Either make it clear that this approach is meant to be a black-box or provide some additional context on why this would converge to physical laws.

Nit: L13 “is only being” can be simply “is only”; “is” defines being.

# Relation to prior work

Very enjoyable and concise coverage of prior work and well connected to the advances laid out in this paper.

# Reproducibility
At a high-level the techniques give a reasonable guide to reproducing the results in this paper. However, invariably, the details are often very important, especially when a statement of performance is made. If the authors are able to provide source-code and working examples this work could truly be reproduced.

**Quality Of The Limitations Section:**

3

**Questions For Rebuttal:**

Please see review for suggestions and nits. I only have one fundamental issue that I would like addressed:

Why is the contact problem formulated via a state-transition? Can the authors please provide prose (math would be wonderful) explaining why the problem wasn't framed in terms of impulse-momentum or energy? And explain why the "average velocity" over time-steps does not violate causality (e.g. present-state has knowledge of future-state) or it doesn't matter?

**Robotics Focus:**

2

**Summary Of Paper:**

This work proposes a method for simulating tensegrity structures using graph-neural networks to aid in modeling contact dynamics. The approach is compared to traditional analytic rigid-body simulation and validated on a tensegrity robot.  The novelty of the work is in the explicit modeling of the tensegrity structure with a graph-neural-network (GNN) to aid in the computation of the contact dynamics, where other attempts either used a generalized formulation (mesh-primitives + GNNs) or relied on rigid-body simulators and first-principles.

**Summary Of Recommendation:**

The paper is a novel application, which provides support for the method of using graph-neural-networks as a technique for modeling complex physical systems. The claims on performance are nascent and would require a solid benchmark paper to be convincing that this technique was truly superior. My pause on giving a solid accept is that I have one lurking doubt about the validity of the problem formulation; classically a position-based euler-scheme would be too stiff and nearly impossible to achieve good results. A solid rebuttal from the authors would convert me to a solid accept.

---

### Author Rebuttal · Authors · 2024-08-10

Contains the revised paper with added information highlighted.

---

### Decision · Program_Chairs · 2024-09-04

**Decision:**

Accept

**Comment:**

PRE REBUTTAL:

High level summary of reviews:

Strengths:

- All reviewers feel that the paper presents a novel application of GNNs to the complex dynamics of tensegrity robots.
- Many reviewers state that the paper is well-written and clear.
- Experiments are thorough (and on real world hardware) and the video included was high quality.
- Improvement over state of the art for tensegrity robots.

Weaknesses:

- Multiple reviewers question whether the experiments generalize well; for example, the models were only tested on flat ground (method assumption) and the real experiments were only conducted on a 3-bar robot.
- Some issues around the problem formulation were raised by multiple reviewers. For example a position-based euler-scheme would be too stiff (Reviewer Luga).
- Additional clarifications on method details were requested (e.g. Mujoco simulation setup, details on MLP, etc). See individual reviewer comments for details.
- The ablation studies are limited and more exploration into different architectures or representations could be beneficial.
- The penetration error is higher and this is not explained in the paper.
- Only incremental improvements on real-world data.

POST REBUTTAL:

The authors provided a very detailed rebuttal and made a number of clarifying updates to the manuscript. Many of the weaknesses raised by the reviewers were resolved. One remaining weakness is still the incremental improvements on real-world data, which will certainly limit the impact of this work within the broader community (as the community is very real-world results driven these days). The authors comment that this will effectively be fixed in another submission (with orthogonal contributions) is understood, but the decision to make these 2 papers rather than one strong paper definitely results in a weaker submission as a result. This paper has a number of novel contributions and is well written is clear, but is ultimately weakened by final results. With that said, the paper marginally crosses the threshold for acceptance. The simulation results might be of interest to the community and the algorithmic contributions are somewhat novel.